# DisenStudio: Customized Multi-subject Text-to-Video Generation with Disentangled Spatial Control

## ABSTRACT

Generating customized content in videos has received increasing attention recently. However, existing works primarily focus on customized text-to-video generation for single subject, suffering from *subject-missing* and *attribute-binding* problems when the video is expected to contain multiple subjects. Furthermore, existing models struggle to assign the desired actions to the corresponding subjects (*action-binding* problem), failing to achieve satisfactory multi-subject generation performance. To tackle the problems, in this paper, we propose DisenStudio, a novel framework that can generate text-guided videos for customized multiple subjects, given few images for each subject. Specifically, DisenStudio enhances a pretrained diffusion-based text-to-video model with our proposed spatial-disentangled cross-attention mechanism to associate each subject with the desired action. Then the model is customized for the multiple subjects with the proposed motion-preserved disentangled finetuning, which involves three tuning strategies: multi-subject co-occurrence tuning, masked single-subject tuning, and multi-subject motion-preserved tuning. The first two strategies guarantee the subject occurrence and preserve their visual attributes, and the third strategy helps the model maintain the temporal motion-generation ability when finetuning on static images. We conduct extensive experiments to demonstrate our proposed DisenStudio significantly outperforms existing methods in various metrics. Additionally, we show that DisenStudio can be used as a powerful tool for various controllable generation applications.

## CCS CONCEPTS

• **Computing methodologies → Computer vision**.

## KEYWORDS

Video Generation, Customization, Multi-Subject, Disentanglement

## 1 INTRODUCTION

On the one hand, with the advent of diffusion models [18, 39, 45], text-to-video generation has witnessed remarkable progress. An increasing number of pretrained text-to-video diffusion models [5, 9, 15, 47, 55] have been proposed recently, enabling users to generate temporally consistent photo-realistic videos by providing the textual descriptions. On the other hand, solely relying on textual prompts cannot fulfill the user's specific customization needs, e.g.,

*ACM MM, 2024, Melbourne, Australia*
© 2024 Copyright held by the owner/author(s). Publication rights licensed to ACM.
ACM ISBN 978-x-xxxx-xxxx-x/YY/MM
https://doi.org/10.1145/nnnnnnn.nnnnnnn

an anime artist may desire to generate a video of their newly created character, or a user may expect to generate videos of their beloved pet dog. It is difficult to determine a textual prompt that can describe all the visual attributes of the anime character or the pet dog. Therefore, customized text-to-video generation [8, 23, 24, 50, 58] has received increasing attention. As shown in Figure 1, given a few images of specific subjects, customized text-to-image generation aims to generate videos that include the subjects and conform to the textual prompts simultaneously.

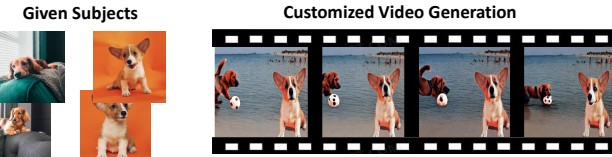

**Figure 1: Illustration for customized multi-subject text-to-video generation.**

However, existing customized text-to-video generation works [24, 50, 58] mostly focus on single-subject customization, limiting their application to broader scenarios of generating videos with multiple customized subjects, e.g., we may want to generate a video of two newly created anime characters dancing together. The single-subject customization works tend to suffer from *subject attribute-binding* (the visual features of different subjects are mixed together) and *subject-missing* (one or more of the multiple subjects is missing) problems when the video is required to contain multiple subjects. Recently, the Disen-Mix finetuning strategy [8] is proposed to distinguish the features of the multiple subjects. However, this work fails to preserve detailed visual attributes of each subject. More importantly, all the existing methods [8, 24, 50, 58] struggle with the action-binding problem. They fail to precisely assign the desired actions to the corresponding subjects as the textual prompts indicate. For instance, when generating a video in which the first dog is kicking a ball and the second dog is sitting, existing methods may generate videos where both of them are kicking the ball or only the second dog is kicking a ball.

To tackle these problems, in this paper we propose DisenStudio, a novel framework capable of generating text-guided videos for customized multiple subjects, given only a few images for each subject. In general, DisenStudio enhances a pretrained diffusion-based text-to-video model with the proposed spatial-disentangled cross-attention, and then conducts customization for the multiple subjects with the proposed motion-preserved disentangled finetuning. Specifically, to tackle the action-binding problem, we propose to replace the vanilla cross-attention in the diffusion model with the spatial-disentangled cross-attention, resulting in the spatial disentanglement of the attention maps for different subject-action pairs.

In addition, to tackle the attribute-binding and subject-missing problem, our proposed motion-preserved disentangled finetuning involves two strategies, i.e., the multi-subject co-occurrence tuning and masked single-subject tuning, where the former utilizes the synthesized multi-subject co-occurrence data during finetuning to avoid subject missing, and the latter guides the model to preserve the visual attributes of each subject with segmentation masks. Moreover, to prevent the text-to-video model from losing temporal motion-generation ability during the finetuning process which only involves static subject images, we propose a novel multi-subject motion-preserved tuning strategy. To further evaluate the performance of different methods, we propose a DisenStudioBench dataset, and conduct extensive experiments on the dataset to show that our proposed method significantly outperforms existing works in various metrics. Our contributions are summarized as follows,

- We propose DisenStudio, a customized multi-subject text-to-video generation framework with disentangled spatial control, allowing for precise assignment of actions to their respective subjects.
- We propose the multi-subject co-occurrence tuning and the masked single-subject tuning strategies to tackle the subject-missing and attribute-binding problem, capable of well preserving the visual attributes of each subject.
- We propose the multi-subject motion-preserved tuning strategy, which maintains the temporal motion-generation ability of the text-to-video model during the process of finetuning.
- We conduct extensive experiments to show that the proposed DisenStudio framework is able to significantly outperform existing baselines in subject fidelity, textual alignment, temporal consistency, and human preference, which can serve as a powerful tool for diverse controllable generation tasks.

## 2 RELATED WORK

*Text-to-image diffusion models.* Text-to-image generation has emerged as a prominent and highly explored topic recently, thanks to the advancements of diffusion models. Trained on large-scale text-image pairs, diffusion models [4, 32, 37–39, 42] can generate photo-realistic images based on textual prompts. GLIDE [32] introduces classifier-free guidance to achieve better text control on images. Dall-E 2 [37] and Imagen [42] leverage pretrained text models to enhance text fidelity. Dall-E 3 tries to improve the generation quality with higher-quality captions. The series of Stable Diffusion (SD) models [12, 34, 39] propose to conduct the diffusion process in the latent space, resulting in improved speed and efficiency while maintaining high-resolution output.

*Text-to-video generation.* The increasing attention on text-to-video generation has been fueled by the success of text-to-image generation. Both diffusion-based models [9, 15, 17, 21, 24, 29, 44, 47, 55, 60] and non-diffusion models [19, 46, 52] have been developed, leveraging pretraining on large-scale video datasets [2, 19, 56]. Compared to the text-to-image diffusion models, temporal modules are designed and pretrained to maintain frame consistency and generate temporal dynamics for text-to-video diffusion models. More recently, a surprising work Sora [5] can even generate 1-minute high-quality videos by applying the transformer structure to the latent diffusion model. Despite the progress made, the general text-to-video generation models still struggle to meet the personalized needs of user-customized subject generation.

*Text-guided video editing.* Text-guided video editing [24, 27, 31, 35, 48, 53, 57, 59] is related to text-to-video generation, which aims to edit the content of a reference video with textual prompts. The difference between text-guided video editing and text-to-video generation is that the former requires an input video while the latter does not. Additionally, it is hard for text-guided video editing to generate new actions. Compared to text-guided video editing, text-to-video generation is a more challenging task.

*Subject customization.* Most subject customization works focus on generating images, which can be categorized into finetuning and non-finetuning methods. Specifically, [10, 13, 14, 16, 26, 41] require finetuning several hundreds of steps on a few reference images of the given subjects, such as DreamBooth [41]. Among these methods, [13, 26, 41] face the attribute-binding problem when applied to multiple subjects. [16] solves the attribute binding problem for multiple subjects by stitching the data but introduces stitching effects. [14] works for a decentralized scenario for multiple subjects. The non-finetuning works [1, 11, 30, 43, 51, 54] use additional datasets to train a visual encoder to provide reference image condition to the generative model, enabling them to customize the subject in a zero-shot manner. Among the non-finetuning methods, [30, 54] consider the multi-subject scenario with attention controls for the attribute-binding problem. However, these non-finetuning methods will fail to customize the subjects that are out-of-domain of the additional datasets, i.e., if the customized subjects do not appear in the additional datasets, they will generate dissimilar subjects.

For customized text-to-video generation, there have been initial attempts [23, 50, 58] that customize the text-to-video models with the reference images of the given subjects. However, these methods are still limited to the single-subject scenario. More recently, Video-Dreamer [8] is proposed to tackle the attribute-binding problem in customized multi-subject text-to-video generation, but it still suffers in preserving the details of each subject. More importantly, it struggles to generate videos where we expect different subjects to take different actions.

## 3 METHODOLOGY

The overall framework of our proposed DisenStudio is shown in Figure 2, which is based on the pretrained AnimateDiff text-to-video generator, whose main component is Stable Diffusion. Therefore, we will first introduce some preliminaries about the Stable Diffusion and AnimateDiff model, and then elaborate on the DisenStudio framework in detail.

### 3.1 Preliminaries

*Stable Diffusion.* Pretrained on large-scale text-image dataset $\{(P, x)\}$, Stable Diffusion [39] can generate photo-realistic images that conform to the text prompts, where $x$ is an image and $P$ is the text description of the image $x$. Different from the pixel-space diffusion model, Stable Diffusion conducts the forward and denoising process in the latent space to improve efficiency, with an encoder $\mathcal{E}(\cdot)$ and a decoder $\mathcal{D}(\cdot)$. The encoder transforms the image $x$ into

Figure 2: The proposed DisenStudio framework is based on the AnimateDiff model that includes the text encoder, and U-Net with temporal modules. Given few images of each subject, (A) we synthesize the multi-subject co-occurrence data with randomly generated background and segmented subjects. (B) we generate images where different subjects take a randomly sampled action, which is used to maintain the motion-generation ability of the model. (C) we finetune the U-Net and text encoder with LoRA, on the synthesized co-occurrence data and generated motion prior data. (D) we insert the temporal modules to U-Net and conduct video generation with the spatial-disentangled cross-attention.

the latent space, $z_0 = \mathcal{E}(x)$, and the decoder reconstructs the image from the latent space with $x \approx \mathcal{D}(z_0)$, where $z_0$ is the latent code. Specifically, in the diffusion forward process, the Gaussian noise is added to the latent code iteratively as follows:

$$q(z_t|z_{t-1}) = \mathcal{N}(z_t; \sqrt{1-\beta_t}z_{t-1}, \beta_t I), t = 1, \cdots, T, \quad (1)$$

where $T$ is large so that $z_T$ is close to a standard Gaussian noise.

In the denoising process, the Stable Diffusion will recover the image latent code $z_0$ from the Gaussian noise $z_T$ step by step. The denoising process relies on a U-Net [40], which we denote as $\epsilon_\theta(\cdot)$, to predict the noise at each step. It receives the noisy latent code $z_t$, timestep $t$, and the textual feature $E_T(P)$ as input, and predicts the noise $\epsilon_\theta(z_t, t, E_T(P))$ at timestep $t$, where $E_T(\cdot)$ is a CLIP text encoder to encode the text prompt $P$. With the predicted noise at each step, we can remove the noise step by step with diffusion samplers [28, 45] until we obtain the clean latent code $z_0$. More specifically, to guarantee that the denoised latent code contains the content described by the prompt $P$, the Stable Diffusion adds cross-attention modules in the U-Net as follows,

$$Attention(Q, K, V) = Softmax(\frac{QK^T}{\sqrt{d}}) \cdot V, \quad (2)$$

$$Q = W_Q \cdot \phi(z), K = W_k \cdot E_T(P), V = W_V \cdot E_T(P),$$

where the text representation $E_T(P)$ will be used as the attention key and value to guide the denoising process. The follow objective is adopted to train the U-Net $\epsilon_\theta(\cdot)$ and the text encoder:

$$\min \mathbb{E}_{P, z_0, \epsilon, t}[||\epsilon - \epsilon_\theta(z_t, t, E_T(P))||_2^2], \quad (3)$$

where for a randomly sampled noise $\epsilon$, we add it to the latent code $z_0$ and obtain the noisy latent $z_t$. What the U-Net $\epsilon_\theta(\cdot)$ needs to do is to make the predicted noise close to the sampled noise $\epsilon$. This objective is widely used during finetuning for customization.

*AnimateDiff.* AnimateDiff [15] is a text-to-video generator based on Stable Diffusion as shown in the left of Figure 2, which adds the green temporal modules to the gray Stable Diffusion text-to-image modules. Specifically, to adapt the original text-to-image Stable Diffusion model to a series of frames of a video, it merges the frame dimension with the original batch size dimension. The merge-dimension operation helps AnimateDiff to utilize the power of Stable Diffusion to process each frame image. Furthermore, to guarantee the generated frames are temporally consistent, Animate-Diff adds a temporal Transformer module after each Stable Diffusion attention block and trains the temporal Transformer modules on the WebVid [3] text-video dataset. Then the model can generate temporally consistent videos with high-quality content from Stable Diffusion prior. In customized multi-subject text-to-video generation, AnimateDiff is very suitable as the base text-to-video generator because it has decoupled image and temporal modules, while we only have few images for each subject. It is very natural to utilize the images to finetune the image module while leaving the temporal module fixed to maintain its motion-generation ability.

### 3.2 A Naive Approach

Based on the preliminaries, a very naive approach for customized multi-subject text-to-video generation is to directly customize the

Stable Diffusion model in AnimateDiff with the given multiple subjects. Specifically, assume that there are $N$ user-defined subjects, and few images for each subject $\{\{x_{ij}\}_{j=1}^{M_i}\}_{i=1}^{N}$, where $x_{ij}$ is the $j^{th}$ image of subject $i$ and $M_i$ (usually 3∼5) is the number of images used for subject $i$. Similar to previous text-to-image customization work [10, 13, 41], we can bind each subject to a special prompt $P_i$ = "a $S_i^*$ $cate_i$" through finetuning, where $S_i^*$ is a rare token to represent the subject identity (e.g., "sks"), and $cate_i$ means the category of subject $i$ (e.g., dog). The finetuning objective is similar to Eq.(3):

$$\mathcal{L}_{naive} = \sum_{i=1}^{N}(\sum_{j=1}^{M_i}\mathbb{E}_{\epsilon,t}[||\epsilon - \epsilon_\theta(z_{ij,t}, t, E_T(P_i))||_2^2]), \quad (4)$$

where $z_{ij,t}$ is the noisy latent code of image $x_{ij}$ at timestep $t$. The inner sum of the objective means given a subject $i$ and its textual prompt $P_i$ as the condition of the U-Net, the U-Net model can denoise for all the images of subject $i$, $\{x_{ij}\}_{j=1}^{M_i}$. Then the text condition $P_i$ is successfully tied to subject $i$. In the outer sum, we will conduct the finetuning process for all the $N$ subjects.

After the finetuning process, we can use the finetuned model and prompts related to $P_i$ to generate new videos as shown in Figure 3, where we use two random seeds to generate two videos and show the frames of the two videos. From Figure 3, we can see that there are two main problems. (i) **Action-binding problem**: the finetuned model fails to assign the desired action to the corresponding subject, e.g., the prompt requires the girl to dance and the dog to play the piano, but in the left frame, the girl plays the piano. (ii) **Attribute-binding and subject-missing problem**: the finetuned model fails to preserve the appearances of the given subjects, e.g., the girl and the dog in the left frame are not so similar to the given two subjects, and in the right frame, the dog is even missing. To tackle the two problems, we propose the DisenStudio framework.

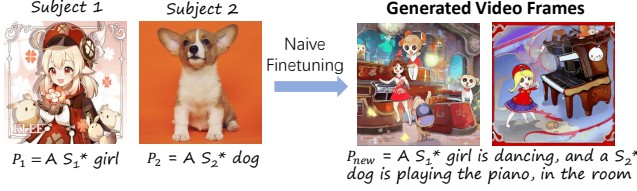

Figure 3: Video frames generated by the naive approach.

### 3.3 DisenStudio

We first focus on the action-binding problem. As indicated by previous works [1, 7, 14], the action-binding problem of Stable Diffusion comes from improper cross-attention results. As shown in Eq.(2), the text representation $E_T(P)$ will attend to all the dimensions of the query $Q$ (latent code), thus making each textual word has the possibility to appear in any place of the generated video frame. Take Figure 3 as an example, the action "playing the piano" can attend to all the regions of the latent code, and thus the attention mechanism makes it possible to attend to the region of the girl instead of the dog. Consequently, the generated frame has the wrong action-binding pattern, "the girl is playing the piano" instead of "the dog is playing the piano" as given in the text. Based on the analysis,

we introduce the following spatial-disentangled cross-attention (SDCA) mechanism to tackle the problem.

*Spatial-disentangled cross-attention.* The difference between the disentangled-spatial cross-attention and vanilla cross-attention is shown in Figure 4. Assuming that $N$ = 2 is the subject number, given a textual prompt $P'$ = "A girl is dancing, and a dog is playing the guitar" that describes the 2 subjects and their actions, we first divide it into 2 prompts, $P'_1$ = "A girl is dancing" and $P'_2$ = "A dog is playing the guitar", which can be easily processed by rules or large language models. After that, we use the two prompts to get two disentangled textual representations, where each representation is related to only one subject and its action. When conducting the cross-attention, the two textual representations will attend to two spatially disentangled regions as follows,

$$Out = [Attention(Q_1, K_1, V_1); \cdots ; Attention(Q_N, K_N, V_N)] \quad (5)$$

$$Attention(Q_i, K_i, V_i) = Softmax(\frac{Q_iK_i^T}{\sqrt{d}}) \cdot V_i, i = 1 \cdots, N$$

$$Q_i = W_Q \cdot \phi(z_i), K_i = W_k \cdot E_T(P'_i), V_i = W_V \cdot E_T(P'_i),$$

where we uniformly divide the original features into N parts, only use one prompt (one subject and its action) to attend to one region, and finally concatenate the N regions together. Therefore, we can make sure that the subjects and their actions are correctly related. Take Figure 4 as an example, "A dog is playing the guitar" will only attend to the right part, so that the action "playing the guitar" will not attend to the region of the girl that is on the left.

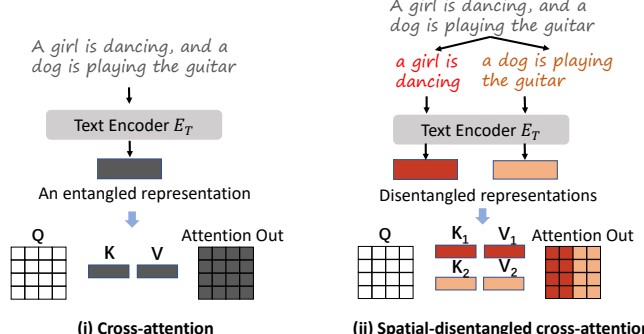

Figure 4: Comparison between the Spatial-disentangled cross-attention and the vanilla cross-attention.

The only problem with the spatial-disentangled cross-attention is whether it will cause discontinuous background because we use different prompts to control different regions. Luckily, thanks to the pretrained knowledge of Stable Diffusion, we find that the images are still continuous as shown in Figure 5 when we apply SDCA to the pretrained Stable Diffusion model. The SDCA tackles the action-binding problem, but we still face the attribute-binding and subject-missing problems. To tackle the two problems, we propose the motion-preserved disentangled finetuning strategy.

*Motion-preserved Disentangled Finetuning.* The motion-preserved disentangled finetuning strategy includes the multi-subject co-occurrence tuning, masked single-subject tuning, and multi-subject

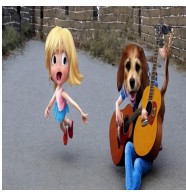 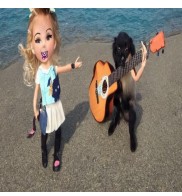

a girl is dancing

a dog is playing the guitar

SDCA

**Figure 5: Generated images from pretrained Stable Diffusion with SDCA are with the continuous background.**

motion-preserved tuning, three components as shown in Part C of Figure 2. We will next elaborate on them in detail.

**Multi-subject co-occurrence tuning.** Since we use SDCA for multi-subject generation, it is consistent for us to adopt SDCA for multi-subject finetuning. However, the images available only contain one subject in each image, preventing us from using SDCA to customize multiple subjects in the same scenario. To tackle the problem, we propose the co-occurrence data synthesizing process as shown in Part A of Figure 2, where we use the segmentation model SAM [25] to segment each subject, and then put them together in the same background generated by the Stable Diffusion. Then we can get a small dataset $D_{mix} = \{x_{mj}\}_{j=1}^{R}$ (3~5 images) where each image contains all the given multiple subjects. With this dataset, we can finetune the Stable Diffusion with the following objective:

$$\mathcal{L}_1 = \sum_{j=1}^{R} \mathbb{E}_{\epsilon,t}[||\epsilon - \epsilon_\theta(z_{mj,t}, t, [E_T(P_1); \cdots ; E_T(P_N)]; SDCA)||_2^2], \quad (6)$$

where we change the original cross-attention to SDCA, and use the $N$ prompts $\{ \text{"a } S_i^* \text{ cate}_i\text{"} \}_{i=1}^{N}$ (e.g., "a $S_1^*$ girl", "a $S_2^*$ dog", "a $S_3^*$ dog" in Figure 2) to respectively attend to the $N$ regions. This finetuning objective ensures that when we use all the special prompts, all the subjects will co-occur in the same frame, avoiding the subject-missing problem.

**Masked single-subject tuning.** To further make each special prompt $P_i$ to preserve the visual attributes of subject $i$, we introduce the masked single-subject customization as follows,

$$\mathcal{L}_2 = \sum_{j=1}^{R} \sum_{i=1}^{N} \mathbb{E}_{\epsilon,t}[||(\epsilon - \epsilon_\theta(z_{mj,t}, t, [E_T(P_i)]; SDCA)) \cdot Mask_i||_2^2], \quad (7)$$

where when we denoise each image in $D_{mix}$, we only denoise one subject with one prompt at a time. Specifically, we use prompt $P_i$ to attend to the $i^{th}$ region, while leaving the other regions conditioned on a NULL prompt, and the denoising loss is masked with the subject mask $Mask_i$ as shown in Part C.(b) of Figure 2, ensuring that using $P_i$ only can generate subject $i$ in the $i^{th}$ masked region, for better preserving each subject's visual details.

**Multi-subject motion-preserved tuning.** Directly using Eq.(6) and Eq.(7) to finetune the Stable Diffusion model can well preserve the attributes of each subject. However, we find it easy to overfit the images in $D_{mix}$, and when we insert the temporal motion module of AnimateDiff, the model will fail to generate videos but a series of static frames, losing the model's motion-generation ability. To tackle the problem, we propose the multi-subject motion-preserved finetuning. As shown Part B of Figure 2, we will first generate some motion prior data with SDCA, using prompt $P_{mot,i} = \{$"a $cate_i$ $action_i$" $\}$, $i = 1, \cdots, N$, where $action_i$ is randomly sampled from

"run, jump, play basketball, walk, play the guitar". Specifically, the motion prior data is obtained as follows,

$$x_{mot,j} = SD(\epsilon_\theta([E_T(P_{mot,1}); \cdots ; E_T(P_{mot,N})]; SDCA)), \quad (8)$$

where we will send $N$ prompts $\{P_{mot,i}\}_{i=1}^{N}$, where each prompt describes one subject of the same category as subject $i$ taking a specific action (e.g., "a girl plays the guitar", "a dog plays the guitar", "a dog walks"), to the Stable Diffusion, and use the Stable Diffusion with SDCA to generate several images with different random seeds. We totally generate 200 images for the motion prior dataset $D_{motion} = \{x_{mot,j}\}_{j=1}^{200}$, and conduct multi-subject motion preserved tuning on the dataset as follows,

$$\mathcal{L}_3 = \mathbb{E}_{\epsilon,t,j}[||\epsilon - \epsilon_\theta(z_{mot,j,t}, t, [E_T(P_{mot,1}); \cdots ; E_T(P_{mot,N})]; SDCA)||_2^2]. \quad (9)$$

With $L_3$, we can preserve the model's ability to generate various motions for different subjects.

**Joint optimization.** The final objective of the motion-preserved disentangled finetuning is: $\mathcal{L} = \mathcal{L}_1 + \mathcal{L}_2 + \mathcal{L}_3$. We follow the previous work [8] to use LoRA [20] to update the text encoder and the U-Net parameters.

*Video generation with disentangled spatial control.* After the finetuning process of the Stable Diffusion, we insert the temporal motion module of the AnimateDiff to generate videos, where we adopt the spatial-disentangled cross-attention to generate multiple subjects with their corresponding actions as shown in Figure 2 D.

## 4 EXPERIMENTS

### 4.1 Experimental Setup

*Dataset.* Since there is no released benchmark for customized multi-subject text-to-video generation, we follow [8] to collect a DisenStudioBench dataset containing 25 subjects, which includes different categories of subjects, such as toys, animation characters, and animals. The images of the datasets are part of the previous works [26, 41], or collected by the authors. In our quantitative experiments, we use 15 multi-subject combinations to evaluate different finetuning methods. The combinations involve 11 2-subject customization, e.g., a dog and an animation girl, and 4 3-subject customization, e.g., an animation girl, a dog, and a cat. During generation, we provide 25 prompts for each customization, involving different actions such as "playing the guitar, playing basketball, sleeping, surfing", different appearances such as "in a red hat", and different backgrounds such as "on the beach, in the flowers". Different from the prompts in [8] where the multiple subjects in a video have the same action, we add prompts that require different subjects to have different actions, such as "a dog is playing the guitar, and a cat is sleeping". We follow the previous work [8] to generate 4 videos with 4 random seeds for each prompt, and obtain 1500 videos for robust evaluation. We provide all the multi-subject videos in the main manuscript and additional results generated by DisenStudio in the supplementary materials.

*Baselines.* We compare our proposed DisenStudio with the recent work VideoDreamer [8] for customized multi-subject text-to-video generation and apply their finetuning method to the AnimateDiff [15] base model. Additionally, we follow VideoDreamer to adopt two customization finetuning methods, i.e., DreamBooth

**Table 1: Quantitative comparisons on DisenStudioBench. ↑ means a larger value indicates better performance and vice versa. The best performance is bolded.**

|  | DINO↑ | CLIP-T↑ | T-Cons↑ | Human-R↓ |
|---|---|---|---|---|
| **DB+AD** | 0.280 | 0.221 | 0.938 | 3.183 |
| **Custom+AD** | 0.289 | 0.236 | 0.911 | 2.978 |
| **VideoDreamer** | 0.362 | 0.225 | 0.948 | 2.403 |
| **DisenStudio** | **0.391** | **0.247** | **0.963** | **1.438** |

(DB) [41] and Customdiffusion (Custom) [26] to finetune the AnimateDiff(AD) model, obtain the DB+AD (i.e., the naive approach) and Custom+AD baselines.

*Evaluation Metrics.* We evaluate all the methods with the following metrics. **DINO** [8, 41]: This metric measures how similar the generated subjects are to the given subjects. We first detect each subject from the generated frames, and calculate the DINO image feature cosine similarity [6] between the detected generated subject and the given subject through version ViT S/16, and finally report the average DINO score on the dataset. A higher DINO score indicates higher similarity to the given subjects and better customization performance. **CLIP-T**: CLIP-T [13, 41] measures whether the generated image conforms to the given textual prompt by the CLIP image and text feature cosine similarity. Here, we calculate the CLIP-T score for each detected subject and their corresponding textual prompt through ViT-L-14 [36]. This metric can reflect whether each subject takes the desired action. **T-Cons**: This metric measures whether the frames are temporally consistent by calculating the CLIP image similarity between frames [53]. **Human-R**: Besides the automatic metrics, we also use human evaluation. Specifically, we asked 40 users of different occupations to rank the videos generated by different methods, by jointly considering whether the generated videos have the same subjects as the given images, whether they are consistent with the text prompts and whether the video is temporally consistent. We randomly sample 10 unique prompts for each user, and we report the average rank, a smaller rank value indicates better performance.

*Implementation.* We implement all the baselines on AnimateDiff [15] with Stable Diffusion v1-5 [39]. Our code is built on the Diffusers library [33]. The finetuning hyper-parameters for DB+AD and Custom+AD are the default parameters provided by the library. We finetune VideoDreamer with their provided settings. In our DisenStudio, the lora rank is 16. The learning rate of the U-Net is 1e-4 and that of the text encoder is 2e-5 as recommended by the Diffusers community. We finetune ~1000 iterations for the customization.

## 4.2 Main Results

*Qualitative results.* We provide qualitative comparisons on the DisenStudioBench dataset in Figure 6 and more are presented in the Appendix. From the results, we can see that both DB+AD and Custom+AD fail to customize the multiple subjects, they either miss one subject or generate subjects with different appearances from the given subjects. Additionally, we find that the generated videos by Custom+AD are often unstable with low temporal consistency,

e.g., the basketball of Custom+AD in Example 1 suddenly appears and changes its color, and in Example 2, the appearance of the dog changes among the video frames. VideoDreamer can better preserve the overall subject appearances than DB+AD and Custom+AD, but some attributes of the subjects may be different. More importantly, it cannot assign the desired actions to the corresponding subject, e.g., in Example 1, the dog generated by VideoDreamer does not wear a yellow scarf and the cat does not play the basketball as the prompt indicates. In contrast, our proposed DisenStudio can best preserve the visual details of each subject and make each subject take their corresponding action. Additionally, the video frames of DisenStudio are clearly consistent as shown in Figure 6.

*Quantitative results.* The quantitative results are provided in Table 1. From the quantitative results, we have the following observations: (i) DisenStudio has the best DINO score, which means the generated subjects are more similar to the given subjects. The superiority is largely due to the multi-subject co-occurrence tuning and masked single-subject tuning, making the special token only focus on the corresponding region and masked subject, thus better preserving the subject visual details. (ii) DisenStudio also has the highest CLIP-T score, meaning that the subjects generated by DisenStudio can better follow the prompt to take their corresponding actions, which benefits from the disentangled spatial cross-attention, making the action only binds to a specific subject. (iii) The frames generated by DisenStudio are more temporally consistent than the baselines. We find that the Custom+AD method is not so stable, often with large changes between consequent frames. Overall, from the human evaluation (Human-R), we find that our generated videos are most favored by humans.

## 4.3 Ablation Studies

In this subsection, we evaluate whether the proposed components are effective, and also some other results generated by DisenStudio.

*Effectiveness of finetuning strategies.* During finetuning, we propose the multi-subject co-occurrence tuning (multi-c), masked single-subject tuning (masked-single), and multi-subject motion-preserved tuning (motion) strategies. We conduct ablations on all the 2-subject combinations for these strategies by respectively removing each of them. The quantitative results are shown in Table 2 and the qualitative results are shown in Figure 7. From the results, we can see that **(i) masked-single&multi-c**: masked single-subject tuning and multi-subject co-occurrence tuning are very important to preserve the visual attributes of each subject. As shown in Figure 7, without multi-c, one subject is missing, and without masked-single, the color of $S_2*$ cat is changed to white from black. Correspondingly, without these two strategies, the DINO score will drop as shown in Table 2. **(ii) motion**: Without the multi-subject motion-preserved tuning, we can see that the CLIP-T score drops, which means it fails to follow the textual prompts and makes the subject take the desired action. More importantly, as we can see from Figure 7 and also the demos we provide in the supplemental material, the generated video w/o motion will be static, and the model overfits the images and loses the motion-generation ability. To further illustrate this problem, we calculate another metric, dynamic degree[22] (abbreviated as Dync) on DisenStudio and w/o motion. This metric

**DB+AD**

**Custom+AD**

Subject 1: A $S_1$* dog

Subject 2: A $S_2$* cat

**Example 1**

**VideoDreamer**

**DisenStudio**

A $S_1$* dog in a yellow scarf is sleeping, and a $S_2$* cat is playing with basketball, on the sofa.

**DB+AD**

**Custom+AD**

Subject 1: A $S_1$* girl

Subject 2: A $S_2$* dog

**Example 2**

**VideoDreamer**

**DisenStudio**

A $S_1$* girl is playing the guitar, and a $S_2$* dog is running, in the flowers.

**Given Subjects**

**Generated Videos**

**Figure 6: Qualitative comparison between DisenStudio and baselines. Baselines suffer from attribute-binding, subject-missing, and action-binding problems. DisenStudio can generate temporally consistent videos that preserve the subject visual details and make each subject take the desired action.**

**Table 2: Ablative quantitative results on the 2-subject combinations of DisenStudioBench, w/o means we remove the corresponding finetuning strategy. Dync is a metric to evaluate whether the videos are dynamic or static, and a smaller Dync means more static videos.**

|  | DINO | CLIP-T | T-Cons | Dync |
|---|---|---|---|---|
| DisenStudio | **0.424** | **0.254** | 0.960 | **0.518** |
| w/o masked-single | 0.409 | **0.254** | 0.961 | - |
| w/o multi-c | 0.386 | 0.246 | 0.950 | - |
| w/o motion | 0.416 | 0.237 | **0.972** | 0.271 |

evaluates whether the generated videos are static, and if Dync is closer to 0, the video is more static, otherwise dynamic, and the mean value of current pretrained text-to-video models [19, 47, 49] is about 0.5. The Dync comparison between DisenStudio and w/o motion shows that multi-subject motion-preserved tuning helps to preserve the motion-generation ability.

*Effectiveness of SDCA.* During generation, we adopt the spatial-disentangled cross-attention (SDCA). To verify its effectiveness, we first replace SDCA with vanilla attention and obtain the variant w/o SDCA. Additionally, we also apply SDCA to our most competitive baseline, VideoDreamer (abbreviated as VD). We still conduct experiments on all the 2-subject combinations of DisenStudioBench. The quantitative results are demonstrated in Table 3 and qualitative results are presented in the Appendix. The results show that without SDCA during generation, DisenStudio will suffer both clear

A $S_1$* cat is walking, and a $S_2$* cat is surfing on board in the ocean, near the beach.

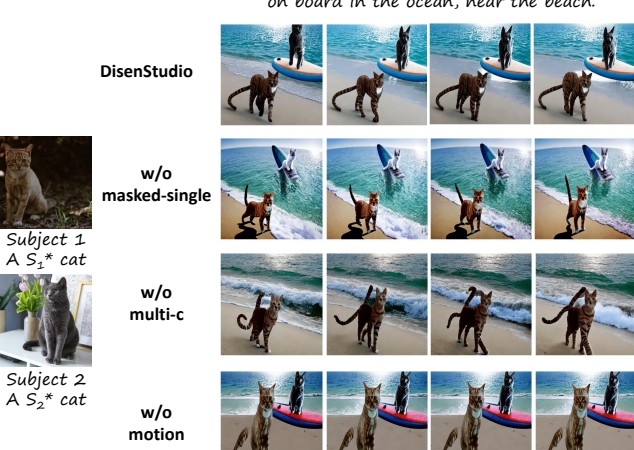

**DisenStudio**

Subject 1
A $S_1$* cat

**w/o masked-single**

Subject 2
A $S_2$* cat

**w/o multi-c**

**w/o motion**

**Figure 7: Qualitative ablation study about the proposed finetuning strategies.**

CLIP-T and DINO drop, because the attributes and actions of the multiple subjects are mixed together in the attention map, making it hard to preserve the subjects' appearances and assign the desired action to the corresponding subject. Additionally, SDCA can also help VideoDreamer to obtain better CLIP-T, but its DINO score is still lower than DisenStudio, further indicating the necessity of the motion-preserved disentangled finetuning.

**Table 3: Ablation about SDCA, where we conduct the experiments on the 2-subject combinations of DisenStudioBench.**

|  | DisenStudio | w/o SDCA | VD | VD+SDCA |
|---|---|---|---|---|
| DINO | **0.424** | 0.386 | 0.392 | 0.395 |
| CLIP-T | **0.254** | 0.240 | 0.224 | 0.232 |
| T-Cons | **0.960** | 0.954 | 0.953 | **0.960** |

*Single-subject generation with spatial control.* During finetuning, we only use the co-occurrence data to customize the multiple subjects. Here, we want to explore whether we can generate videos for a single subject. Furthermore, since SDCA is related to the region of each subject, we want to explore whether we can use the SDCA to control the position of the subject. The results are provided in Figure 8, and it demonstrates that our method can be also applied to generating videos about any of the multiple subjects, and provide detailed control of their location in the videos.

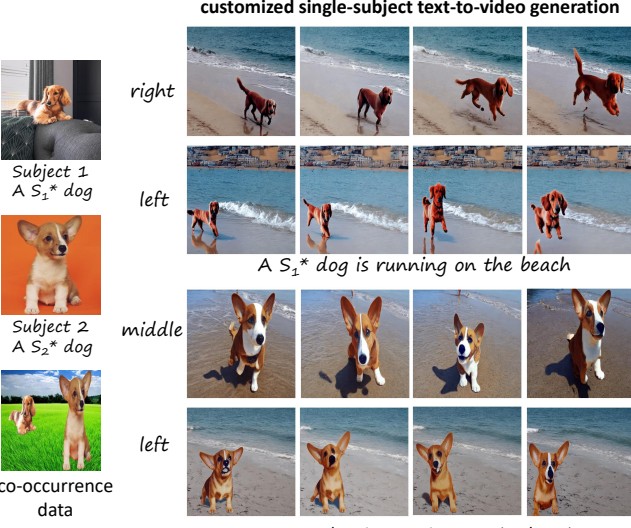

**Figure 8: Customized single-subject text-to-video generation with SDCA to control the subject position.**

*DisenStudio as a video storyteller.* Since DisenStudio can generate videos for both single and multiple subjects, it is easy to use DisenStudio to create several video stories about these subjects. Here, we show a simple example in Figure 9, where we imagine a story that "*the girl is playing the guitar, and the guitar music attracts the two dogs. Then the two dogs run to her, and sit beside the girl to listen to the guitar music.*" The prompts used to generate each video are respectively: "*A $S_1*$ girl is playing the guitar in the flowers*", "*A $S_2*$ dog and a $S_3*$ dog are running in the flowers*", "*A $S_1*$ girl is playing the guitar, a $S_2*$ dog is sitting, and a $S_3*$ dog is sitting in the flowers*". We believe DisenStudio will boost more interesting applications.

*More generation results.* Previously, all the generation results are about subjects in different regions. Here, we want to explore

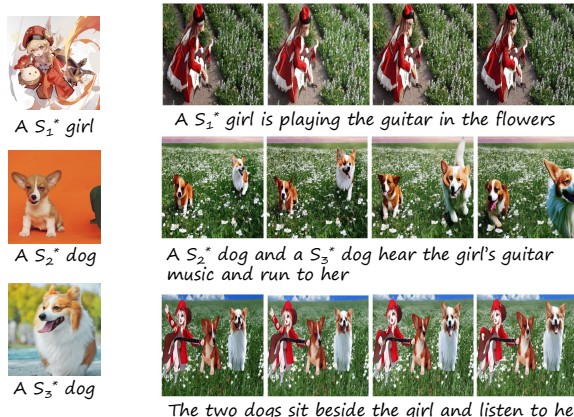

**Figure 9: A simple example to show DisenStudio can be used as a video storyteller.**

whether our method can generate interactions between different subjects. By placing the cross-attention regions of different subjects at different relative positions in the entire frame, we obtain the results in Figure 10. We can see that DisenStudio can also generate interactions such as "a girl holding or riding a dog". When generating "riding", we put the attention region of the girl on top of the attention region of the dog. When generating "holding", we put the attention region of the dog inside the attention region of the girl.

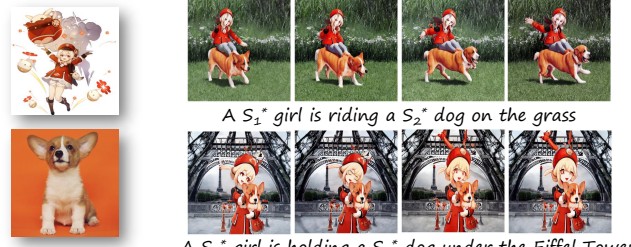

**Figure 10: DisenStudio generates multi-subject interactions.**

## 5 CONCLUSION

In this paper, we propose a DisenStudio framework for customized multi-subject text-to-video generation. We propose the spatial-disentangled cross-attention for generation to tackle the action-binding problem. Additionally, we propose the motion-preserved disentangled finetuning which involves three tuning strategies: multi-subject co-occurrence tuning and masked single-subject tuning to tackle the attribute-binding problem, multi-subject motion-preserved tuning to preserve the model's motion-generation ability. Extensive experimental results demonstrate that our proposed DisenStudio significantly outperforms existing works, and DisenStudio can work as a powerful for various applications. Future works can consider applying DisenStudio to more advanced base text-to-video generators and combining DisenStudio with other controllable generation methods such as ControlNet.

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
