# OpenReview forum: "DisenStudio: Customized Multi-subject Text-to-Video Generation with Disentangled Spatial Control"
_acmmm.org/ACMMM/2024/Conference — MM2024 Poster_

### Official Review · Reviewer_Jh63 · 2024-05-26

**Rating:** 4
**Confidence:** 2

**Summary:**

This paper aims at the text-to-video generation task. Previous methods assign the desired actions to the corresponding subjects, which might obtain unsatisfactory performance. Authors present the spatial-disentangled cross-attention to deal with the action-binding problem. Extensive experimental results show the effectiveness of the presented method.

**Strengths:**

This paper is easy to follow. The presented DisenStudio seems novel.

The topic is interesting and the figures are clear.

Qualitative experiments are sufficient for me.

**Limitations:**

Authors only report the effectiveness comparison. What about the efficiency comparison and computational cost?

Will authors make their code publicly available?

Which GPU did authors use? How long to train and test?

Author should provide more details about the implementation because the paragraph is too short.

**Suitability:**

2

---

### Official Review · Reviewer_B9F3 · 2024-05-26

**Rating:** 4
**Confidence:** 3

**Summary:**

- The paper proposes a framework for customized multi-subject text-to-video generation by adopting a disentangled spatial control approach. The proposed method addresses the problems of subject-missing and attribute-binding that exist in current works, which mainly focus on generating single-subject content. The proposed DisenStudio framework involves three tuning strategies: multi-subject cooccurrence tuning, masked single-subject tuning, and multi-subject motion-preserved tuning. And the experiments demonstrate the effectiveness of the proposed framework.

**Strengths:**

- The proposed method in this paper is practical and reasonable throughout the entire pipeline from data to model. The three tuning strategies have technical applicability. Additionally, the authors provide extensive visualizations of the generated results, which helps to intuitively analyze and understand the application effects of the proposed method.

**Limitations:**

- I think the main weaknesses of this paper are the lack of novelty and the limited contribution to the field. For the complex task of multi-subject text-to-video generation, the paper simply categorizes it into three types of tasks and realizes three tuning strategies. The overall framework is relatively simple without introducing many novel techniques. Additionally, the scale of the task and the amount of data involved in this work are relatively small, which might provide a limited contribution to the research field. Noting that, I am not very specialized in this field. I expect the authors to analyze and discuss the weakness I raise in the rebuttal, and I will also take into account the opinions of other reviewers before giving a final decision. My current rating is borderline accept.

**Suitability:**

3

---

### Official Review · Reviewer_ME4K · 2024-06-03

**Rating:** 5
**Confidence:** 3

**Summary:**

This paper introduces the DisenStudio framework for customized multi-subject text-to-video generation, focusing primarily on addressing attribute-binding and subject-missing challenges. It proposes three key tuning strategies. Specifically, the motion-preserved disentangled finetuning strategy is presented to tackle these challenges. This strategy employs multi-subject co-occurrence tuning and masked single-subject tuning to ensure the appearance of each subject, while multi-subject motion-preserved tuning maintains the model's motion-generation ability. To enhance the accuracy of assigning different actions to various subjects, the authors suggest adopting disentangled-spatial cross-attention instead of the vanilla cross-attention mechanism. Extensive experiments validate the effectiveness of the proposed approach.

**Strengths:**

1. The paper is clearly written and well-motivated, focusing on an aspect that existing text-to-video works generally ignores, i.e., generating multiples subjects and assigning different actions to different subjects.
2. The experimental results demonstrate that DisenStudio show superior customized multi-subject text-to-video generation results, effectively tackling the attribute-binding and action-binding problems. Additional experiments show that the proposed method has potential for more controllable generation applications.

**Limitations:**

1. As indicated in the appendix of the paper, one limitation is from the adopted foundation text-to-video model, which only supports short video-clip generation. I understand this is due to the limitation of current open-source models, but I hope the authors could apply the customization methods to more advanced models, like videocrafter.
2. The authors focus on the Action-binding problem, Attribute-binding, and subject-missing problem. Therefore, it would be appropriate to compare with some T2I models that address these issues, such as attend-and-excite[1]+AD,  Linguistic binding[2]+AD, or Divide-and-bind[3]+AD.
3. From the qualitative results, it can be seen that there is still a varying degree of subject distortion in the generated results. The authors need to provide an explanation for this phenomenon.

[1] Chefer H, Alaluf Y, Vinker Y, et al. Attend-and-excite: Attention-based semantic guidance for text-to-image diffusion models[J]. ACM Transactions on Graphics (TOG), 2023, 42(4): 1-10.
[2] Rassin R, Hirsch E, Glickman D, et al. Linguistic binding in diffusion models: Enhancing attribute correspondence through attention map alignment[J]. Advances in Neural Information Processing Systems, 2024, 36.
[3] Li Y, Keuper M, Zhang D, et al. Divide & bind your attention for improved generative semantic nursing[J]. arXiv preprint arXiv:2307.10864, 2023.

**Suitability:**

3

---

### Meta-Review · Area_Chair_c7eJ · 2024-07-01

**Recommendation:** Accept (Poster)
**Confidence:** 5

**Metareview:**

This paper introduces a framework for customized multi-subject text-to-video generation. It is clearly written, aiming to facilitate understanding of the proposed three key tuning strategies.  The experiments are comprehensive, and the visualizations of the generated results are commendable. However, the overall framework is relatively simple with limited novelty and contribution to the field, and it lacks some details. In summary, all review opinions were unanimous in favor of acceptance with aware of limitation of this paper.